# The Path of a Cardiac Patient—From the First Symptoms to Diagnosis to Treatment: Experiences from the Tertiary Care Center in Poland

**DOI:** 10.3390/jcm11185276

**Published:** 2022-09-07

**Authors:** Przemysław Seweryn Kasiak, Barbara Buchalska, Weronika Kowalczyk, Krzysztof Wyszomirski, Bartosz Krzowski, Marcin Grabowski, Paweł Balsam

**Affiliations:** 1st Chair and Department of Cardiology, Medical University of Warsaw, Banacha 1A, 02-097 Warsaw, Poland

**Keywords:** cardiologic care, cardiovascular disease, diagnosis, quality of care, treatment

## Abstract

Cardiovascular diseases (CVDs) are major concerns in the healthcare system. An individual diagnostic approach and personalized therapy are key areas of an effective therapeutic process. The major aims of this study were: (1) to assess leading patient problems related to symptoms, diagnosis, and treatment of CVDs, (2) to examine patients’ opinions about the healthcare system in Poland, and (3) to provide a proposal of practical solutions. The 27-point author’s questionnaire was distributed in the Cardiology Department of the Tertiary Care Centre between 2nd September–13th November 2021. A total of 132 patients were recruited, and 82 (62.12%; n_male_ = 37, 45.12%; n_female_ = 45, 54.88%) was finally included. The most common CVDs were arrhythmias and hypertension (both *n* = 43, 52.44%). 23 (28.05%) patients had an online appointment. Of the patients, 66 (80.49%) positively assessed and obtained treatment, while 11 (13.41%) patients declared they received a missed therapy. The participants identified: (1) waiting time (*n* = 31; 37.80%), (2) diagnostic process (*n* = 18; 21.95%), and (3) high price with limited availability of drugs (*n* = 12; 14.63%) as the areas that needed the strongest improvement. Younger patients more often negatively assessed doctor visits (30–40 yr.; *p* = 0.02) and hospital interventions (40–50 yr.; *p* = 0.008). Older patients (50–60 years old) less often negatively assessed the therapeutic process (*p* = 0.01). The knowledge of the factors determining patient adherence to treatment and satisfaction by Medical Professionals is crucial in providing effective treatment. Areas that require the strongest improvement are: (1) waiting time for an appointment and diagnosis, (2) limited availability and price of drugs, and (3) prolonged, complicated diagnostic process. Providing practical solutions is a crucial aspect of improving CVDs therapy.

## 1. Introduction

Cardiovascular diseases (CVDs), primarily ischemic heart disease and stroke, are one of the leading causes of death worldwide [1,2]. Despite numerous efforts, the prevalence and incidence of CVDs are still rising, especially in low- and middle-income countries [1]. Hence, it is of the highest importance to develop new, effective methods of treatment and implement them in health care systems [3]. If the introduced changes should be effective and respond to the patient’s needs, it is crucial to acknowledge physicians with the current requirements and the areas that need improvement [3]. As part of the patients’ involvement, it is also worth knowing what they pay attention to during their hospitalization and appointments. It will further facilitate physicians’ cooperation with patients and provide more personalized therapy.

Individualized treatment and diagnostic approaches are essential elements of effective therapy [2,4]. It is especially important in CVD management, as CVDs affect all ages and social groups, and numerous diseases could be treated in outpatient circumstances [1,5]. A correctly implemented treatment protocol facilitates the development of patient compliance, which remains the main pillar of the cardiological care [4,5]. Developing the patient’s voluntary rigor (i.e., regularity in taking medications, measuring the blood pressure, heart rate, and other vital functions, as implementing proper lifestyle changes) and compliance with medical recommendations reduces the overload on particular hospital departments [5]. 

Currently, many variables are described as negatively affecting adherence to recommendations provided by medical professionals [6]. The main predictors are lack of understanding of the treatment protocol and goals of the therapy, insufficient patient health education, and factors related to the limited availability of drugs, their high price, and long waiting times for appointments [7]. However, the problem with developing unified compliance recommendations is the constantly changing attitude of patients, their expectations, and areas of healthcare where improvement is required [8,9]. Hence, providing comprehensive reports and collecting therapy outcomes in hospitals of all levels of specialty (including primary, secondary, and tertiary care Centers) is a crucial element in increasing the effectiveness of the health care system.

We assume that the improvement of the effectiveness of the health care system, and in particular the field of cardiology, will increase patient satisfaction with the therapy and have a positive impact on their compliance.

The aims of this study were: (1) to assess patients’ opinions about the recommendations they receive from their attending physicians, (2) to recognize the steps the patients are taking to obtain a diagnosis of their symptoms, and (3) to identify the main ways of knowledge that patients use for self-education about their CVDs, (4) to point out major areas that require improvement, and (5) provide the direction of potential changes in the healthcare sector.

## 2. Materials and Methods

### 2.1. General Study Design and Data Collection Process

The questionnaire was fulfilled by 132 patients from the cardiology department at the tertiary care diagnostic center (University Clinical Center of the Medical University of Warsaw; https://uckwum.pl/, accessed on 17 July 2022). Data were collected at (1) the Clinical Department of General Cardiology, and (2) the Department of Intensive Cardiac Care. The clinic consists of 4 sub-departments and offers a wide spectrum of diagnostics and treatment, from basic procedures (Echocardiography), through more specialized (Cardiac ablation) to highly advanced (TricValve^®^; P+F Products & Features GMBH, Wessling, Germany). The inclusion criteria were: (1) admission to the hospital at the cardiology department, (2) answering all questions (no empty fields). In order to maximize the credibility of the analyzed data and to exclude people with unviable and lacking answers (with a high risk of misunderstanding the survey and study assumptions), the data-cleaning process was applied. All participants met criterion number 1. Patients who did not meet criterion number 2 were excluded from further analysis (*n* = 50; 37.88%). A total of 82 patients (62.12%) met all inclusion criteria. Data were collected from 2 September 2021 to 13 November 2021. The patient’s name and the room they were staying in the clinic were noted during hospital admission (only clinicians know the patients’ data). This enabled verification of patients and ensured that no one completed the questionnaire more than once, but also that all patients fulfilled the form. The questionnaire did not include the question for name or surname; therefore, it was fully anonymous. Patients received a questionnaire during their hospital admission. Data were obtained via in-person meetings with the usage of the paper survey or via the online form. The participant could receive a link to an interactive questionnaire and complete it during the hospital stay from any device at any time. The terms of participation in the study and data anonymity regulations were described at the beginning of the form. By completing the survey, participants gave their informed consent to participate in the study. Participation in the study was fully voluntary. Patients did not receive any financial or material benefits for completing the questionnaire. According to the regulations of the Bioethics Committee of the Medical University of Warsaw, the study did not require registration and further consent.

### 2.2. Construction of the Questionnaire

The 27-point questionnaire was prepared and jointly agreed upon by experts and physicians from the hospital’s cardiology clinic. The survey consisted of the author’s original questions related to (1) demographic data of participants, (2) past medical history, (3) diagnostic process, (4) current medical conditions and therapeutic process, (5) personal thoughts about the disease and health care sector. The survey consisted of two types of questions—(1) closed (*n* = 17) and (2) open (*n* = 10)—in which patients could provide their own answers. In the last two questions, participants could express their own thoughts about the disease and feelings related to the therapeutic process or health care system functioning in Poland. The original questionnaire form is available in the printed English version at the Appendix A and in the Polish online version via the link (https://docs.google.com/forms/d/e/1FAIpQLSf97PdpxCVIraD_ZWgNMabAR8kRbDPacouAbqO3zUoxuRqtKg/formResponse; accessed on 17 July 2022).

### 2.3. Data Analysis

The data were exported to the Excel spreadsheet (Microsoft Corporation, Redmond, WA, USA). Statistical analysis was performed in the STATISTICA software (version 13.3, StatSoft Polska Sp. z o.o., Kraków, Poland) and SPSS software (version 28; IBM SPSS, Chicago, IL, USA). Basic calculations were made, and categorical data were calculated as numbers (*n*) with percentages (%). General linear models and one-way ANOVA [10,11] were applied to assess correlations between clinical and demographic variables and were presented in accordance with the unified APA guidelines [12]. The results were additionally presented with the usage of 95% confidence intervals (CI). The borderline for statistically significant results was defined as *p*-value = 0.05. Graphical abstract was created with BioRender.com (https://biorender.com/, accessed on 26 July 2022; BioRender, Toronto, ON, Canada).

## 3. Results

### 3.1. Study Group Characteristic

We collected surveys from 82 patients. Of the patients, 37 (45.12%) were females and 45 (54.88%) were males. The majority of the patients (*n* = 57; 69.51%) were above 60 years old, while 13 (15.85%) patients were between 50–60 years old, 7 (8.54%) were 40–50 years old, and 5 (6.10%) individuals were 30–40 years old. The most common conditions and complexes the patients were diagnosed with depending on their age are presented in Table 1. Briefly, the most frequent were arrhythmias (*n* = 43; 52.44%) and hypertension (*n* = 43; 52.44%). Table 2 presents the symptoms experienced by the patients stratified by age. The most frequently reported symptom was dyspnea (*n* = 26; 31.71%). A total of 38 (46.34%) patients were reading about their symptoms on the Internet. A total of 62 (75.61%) patients had diagnostic tests. The diagnostic tests were reimbursed to the majority of the patients (*n* = 63; 76.83%), and 17 (20.73%) of them had private health insurance. A total of 61 (74.39%) patients had an attending physician. The first step in the diagnostic investigation was an examination by the physician (*n* = 45; 54.88%) as presented in Table 3. Most of the patients had control appointments, which usually occurred every 3 months (*n* = 38; 46.34%). Only 23 (28.05%) patients had an online appointment with a cardiologist (*n* = 6; 7.32% had a paid fee for an online appointment). The majority of patients (*n* = 66; 80.49%) felt “taken care of” at the hospital. In total, 33 (40.24%) patients reported that cardiac disease negatively affects their daily living. Only 11 (13.41%) patients said that the therapies they received were missed. The most frustrating elements in the diagnostic process were the appointments with the doctors (*n* = 31; 37.81%), medical tests (*n* = 18; 21.95%), and the purchase of medications (*n* = 12; 14.63%). Figure 1 shows duration of the diagnostic investigations.

### 3.2. Clinical Characteristic

The females more commonly received the diagnosis of hypothyroidism (*p* = 0.02, F[1, 63] = 5.49), and males more frequently received the diagnosis of valvular heart disease (*p* = 0.04, F[1, 63] = 4.35). Younger patients (30–40 years old) pointed out that the appointments with the doctor were the most frustrating elements in the diagnostic process (*p* = 0.02, F[1, 61] = 5.83). Furthermore, they more often bought the drugs on the Internet or did not buy any drugs at all, rather than buying drugs at the pharmacy (*p* = 0.04, F[1, 61] = 4.38). Patients aged 40–50 years rated the hospital interventions as the most frustrating (*p* = 0.008, F[1, 61] = 7.54). Patients aged 50–60 years less frequently had atherosclerosis than other conditions (*p* = 0.03, F[1, 62] = 5.06). However, the oldest patients (above 60 years of age) more commonly were diagnosed with atherosclerosis than with other conditions (*p* = 0.0007, F[1, 62] = 12.68), and rated the medical tests less frustrating (*p* = 0.01, F[1, 61] = 7.08). For further analysis related to the clinical characteristics of participants see Figure 2.

### 3.3. Open Questions

The questionnaire also contained open questions numbers 26 and 27. Patients could express their own opinions and add commentaries about CVD-related lifestyle restrictions and online appointments. Descriptive responses acquired from each patient are presented in Appendix A. Briefly, participants mostly reported the negative impact of their CVD on numerous lifestyle areas, indicating worsening workability, and a decrease in physical fitness. Individuals also declared that daily activities such as shopping or household chores are more difficult for them. Patients underlined that they prefer stationary visits to online methods. They indicated the possibility of performing a wider spectrum of diagnostic tests and direct contact with their attending physician as a major advantage of in-person appointments. Respondents expressed their negative thoughts about the medical care system in Poland, pointing out its ineffectiveness, long waiting times, and lack of receiving proper treatment recommendations. The answers varied in characteristics and length, from single comments to multi-sentence statements. A minority of the respondents claimed a positive outcome, mostly expressing gratitude to medical professionals for their work. 

## 4. Discussion

In this study, we present the variable opinions of patients from a highly specialized cardiology clinic at a tertiary care center in Poland. The unquestionable advantage of this study is the protocol of data collection. All questionnaires were obtained from registered and other individuals. Data were provided during in-person meetings or in an online-based controlled setting, which maximized the credibility of the received responses. It allowed for reliable conclusions, and the data collected in this way provide valuable material for the preparation of practical solutions and recommendations.

The novelty of this study is also its comprehensive approach because our survey covers various stages of cardiac care, from the occurrence of the earliest symptoms, by obtaining a definitive diagnosis, to the undergoing full treatment process. We also examined additional patients’ opinions on medical education, their sources of health knowledge, attitudes to attending physicians, etc.

Finally, we prepared a set of practical recommendations and solutions, the implementation of which should increase the effectiveness of cardiological care, and thus positively impact the patient’s compliance.

The study population was mostly the elderly, above 60 years of age, which is commonly seen in the case of CVDs [13]. This implies that diagnostic investigations should be specifically accustomed to older patients so that they will receive the proper treatment.

The majority declared they felt “taken care of”, which is one of the indicators of receiving good healthcare and proper diagnostic procedures [14]. Our results are in line with those provided by Deaton et al., because in other hospitals CVD patients also are satisfied with the amount of care they receive [15].

However, as much as 40% of our patients reported that they have a decreased quality of life due to CVD, which negatively affected their daily living. Unfortunately, a decreased quality of life is commonly seen in people with CVDs [16,17]. Thus, we explored one of the areas where special efforts have to be made in improving the well-being of the patients. Perhaps that could be achieved by additional psychological and social care [18], as CVD is associated with numerous limitations in variable lifestyle areas such as occupational abilities [19], and these restrictions strongly affect mental health as well.

As the most frequently reported constraints were worsening workability, and a decrease in physical fitness, the patients should receive the appropriate rehabilitation after the treatment to overcome the inconveniences [20,21]. We propose simple solutions that could be considered by employers to include: (1) modification of the work mode, e.g., by limiting night or unbroken shifts, and (2) extending the number of vacation/rest days for patients with CVD. There is also a wide field for the application of medical rehabilitation and fitness training [21,22]. During visits and at discharge from the hospital, patients should receive personalized recommendations from their attending physicians regarding physical activity, its amount (i.e., number of sessions per week), form (i.e., strength or cardio training, yoga, etc.), and intensity (based on “speech test”, percentage of maximal heart rate, oxygen uptake, or subjective feeling) [23]. A similar solution has already been introduced at the University Clinical Center of the Medical University of Warsaw, referred to as Managed Care after Acute Myocardial Infarction (“KOS-zawal”). Wita et al. found that cardiac post-infarction rehabilitation can reduce mortality by as much as 45%. Moreover, the data suggest that patients are satisfied with such a treatment protocol despite it being implemented only as part of outpatient hospital care [24].

The patients also highlighted the negatives associated with healthcare in Poland such as ineffectiveness, long waiting times, and lack of receiving proper treatment recommendations. This may be due to a lack of human resources, shortened appointment time spent for discussing and explaining doubts, as well as the lack of funding for the hospital sector [25,26]. Government investments [27] in medical education (e.g., by increasing the number of universities, places of internships, and improving the current education environment) would allow not only an increase in the number of graduates but also primarily increase the number of specialist doctors [28,29]. The general practitioner is responsible for screening tests and long-term monitoring of a CVD patient [30,31], whereas the specialist is responsible for definitive diagnosis and prescribing advanced, patient-centered, and individualized treatment [32,33]. In an effective cardiac care scheme, the role of both specialists—general practitioners and specialists, and their collaboration is crucial. What is more, general practitioners in Poland are overworked. Hence, it is difficult for them to conduct effective screening tests, and therefore patients are later admitted to a specialized diagnostic center. Thus, they are presented with more advanced conditions [34].

To summarize, all these factors contribute to the lengthened time needed to make a proper diagnosis and, perhaps, could provide missed therapies. Consequently, the above-described variables lower the patient’s compliance, their tendency to trust the medical professionals, and let them provide and control a comprehensive therapeutic process [6,35]. Moreover, those factors favor the patient’s search for alternative and faster methods of treatment which often are not supported by evidence-based medicine and derive from beliefs and subjective feelings [35].

### 4.1. Further Studies Directions

As this study was the first use of the questionnaire, it has not yet been externally validated in other populations. To improve the precision and accuracy of the quality of life assessment for cardiologic patients, we recommend further studies which apply our questionnaire for varied populations (both healthy and clinical), and perhaps pair it with other well-studied, validated quality of life questionnaires (i.e., WHOQOL-100 [36] and the WHOQOL-BREF [37]).

As our study was conducted at a tertiary care hospital in Poland and this study is single-center experience, the results will be most valuable locally and regionally to improve the quality of care provided in Poland. We recommend that further studies should compare our questionnaire and similar forms (i.e., CMS-mandated HCAHPS survey [38]) to investigate its transferability.

### 4.2. Limitations

Some limitations should be mentioned when analyzing the results. The study group is primarily small, and further populational studies on wider samples should be conducted (perhaps including patients from local hospitals to enrich sample variability). The incidence of certain diseases (e.g., arrhythmias) may differ from the general characteristics due to the specialization profile of the hospital. Moreover, the declared diseases of the patients have not been verified with the actually diagnosed ones in their medical documentation. Hence, a few inaccuracies may occur as patients are not always able to accurately define their condition [39]. We did not ask patients about their economic status and individuals with higher salaries could describe the treatment and particular procedures as expensive in other conditions than those with lower income. Thus, our outcomes have to be interpreted carefully. Due to the data’s self-reported characteristics, they could be subjective. To minimize the impact of the above-mentioned limitations, we applied an additional data cleaning protocol and provided precise instructions for each part of the questionnaire. Moreover, we recommend further studies of similar protocols on variable and wide populations at all levels of cardiac care.

## 5. Conclusions

CVD patients assessed the effectiveness of the cardiac care system in Poland as moderate with numerous areas requiring improvement. Despite, the majority being declared “taken care of”, younger participants often reported negative outcomes. The areas that were most commonly indicated as needed improvement were the availability and price of drugs, as well as waiting time for and the quality of medical appointments. Knowledge about the current situation and patient opinions provides valuable information for medical professionals and should be used in the development of long-term programs to increase the effectiveness of healthcare systems.

## Figures and Tables

**Figure 1 jcm-11-05276-f001:**
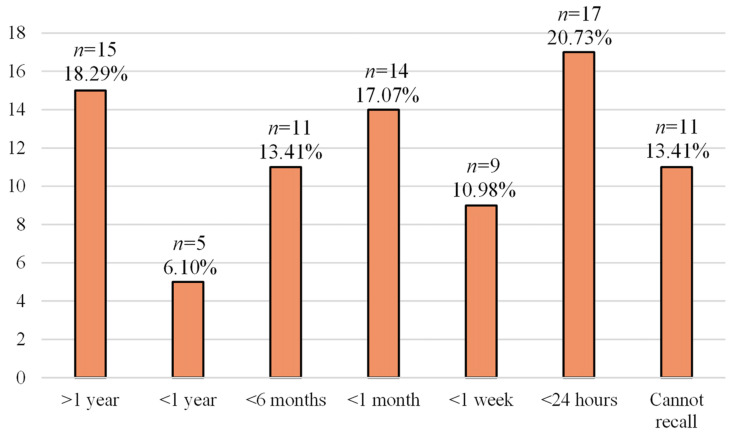
The time in which the patients obtained definitive diagnosis. Data are presented as the number of patients and the percentage of the whole population.

**Figure 2 jcm-11-05276-f002:**
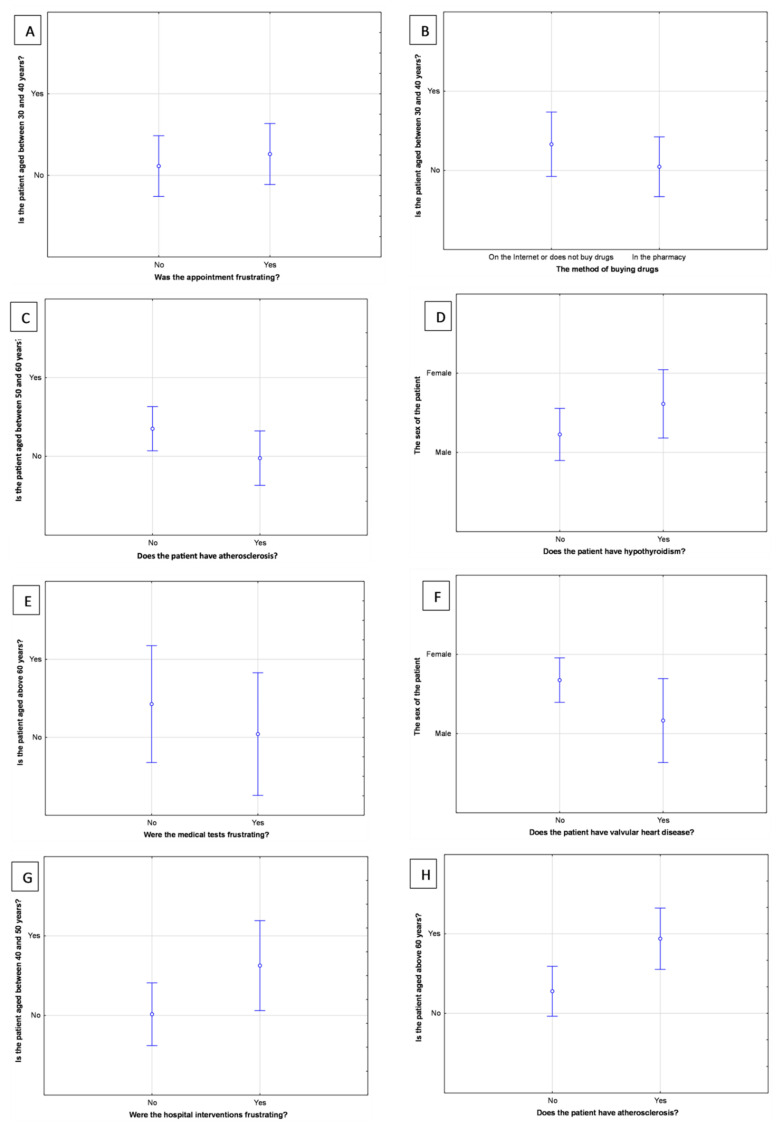
The results of general linear models. (**A**) The frustration of the appointment at the doctor in the group of patients aged between 30 and 40 years. (**B**) The method of buying the drugs in the group of patients aged between 30 and 40 years. (**C**) The occurrence of atherosclerosis in the group of patients aged between 50 and 60 years. (**D**) The occurrence of hypothyroidism in males and females. (**E**) The frustration of the medical tests in the group of patients aged above 60 years. (**F**) The occurrence of valvular heart disease in males and females. (**G**) The frustration of the hospital interventions in the group of patients aged between 40 and 50 years. (**H**) The occurrence of atherosclerosis in the group of patients aged above 60 years. Panels show box-and-whiskers plots. The panels present the statistical relationship between particular variables at the X and Y axes. The longer the whiskers are and the more centrally the median point is located between variables, the weaker correlation between the two variables presented at the X and Y axes.

**Table 1 jcm-11-05276-t001:** Conditions and complexes which the patients were diagnosed with. Data are additionally stratified by age and presented as the number of patients with a percentage of the whole population or a particular subgroup.

Condition/Complex	Whole Population		30–40 Years		40–50 Years		50–60 Years		>60 Years	
	*n* of Patients	% of the Group	*n* of Patients	% of the Subgroup	*n* of Patients	% of the Subgroup	*n* of Patients	% of the Subgroup	*n* of Patients	% of the Subgroup
Arrythmias	43	(52.44%)	3	(6.98%)	4	(9.30%)	5	(11.63%)	31	(72.09%)
Hypertension	43	(52.44%)	3	(6.98%)	4	(9.30%)	8	(18.60%)	28	(65.12%)
Overweight	25	(30.49%)	1	(4.00%)	4	(16.00%)	6	(24.00%)	14	(56.00%)
Type 2 diabetes mellitus	24	(29.27%)	1	(4.17%)	2	(8.33%)	6	(25.00%)	15	(62.50%)
Coronary artery disease	23	(28.05%)	0	(0.00%)	0	(0.00%)	5	(21.74%)	18	(78.26%)
Heart failure	17	(20.73%)	2	(11.76%)	1	(5.88%)	3	(17.65%)	11	(64.71%)
Hypercholesterolemia	14	(17.07%)	0	(0.00%)	3	(21.43%)	2	(14.29%)	9	(64.29%)
Chronic pulmonary disease	13	(15.85%)	0	(0.00%)	2	(15.38%)	1	−7.69%	10	−76.92%
Hypothyroidism	13	(15.85%)	0	(0.00%)	3	(23.08%)	2	(15.38%)	8	(61.54%)
Depression	10	(12.20%)	2	(20.00%)	1	(10.00%)	2	(20.00%)	5	(50.00%)
Atherosclerosis	9	(10.98%)	0	(0.00%)	0	(0.00%)	0	(0.00%)	9	(100.00%)
Valvular heart disease	5	(6.10%)	0	(0.00%)	2	(40.00%)	1	(20.00%)	2	(40.00%)

**Table 2 jcm-11-05276-t002:** Symptoms reported by the patients during the diagnostic process. Data are additionally stratified by age and presented as the number of patients with a percentage of the whole population or a particular subgroup.

Symptom	Whole Population		30–40 Years		40–50 Years		50–60 Years		>60 Years	
	*n* of Patients	% of the Group	*n* of Patients	% of the Subgroup	*n* of Patients	% of the Subgroup	*n* of Patients	% of the Subgroup	*n* of Patients	% of the Subgroup
Dyspnea	26	(31.71%)	2	(7.69%)	2	(7.69%)	5	(19.23%)	17	(65.38%)
Pain in the chest	20	(24.39%)	1	(5.00%)	2	(10.00%)	2	(10.00%)	15	(75.00%)
Exertion fatigue	20	(23.17%)	3	(15.00%)	2	(10.00%)	2	(10.00%)	13	(65.00%)
Palpitations	19	(20.73%)	2	(10.53%)	1	(5.26%)	2	(10.53%)	14	(73.68%)
Tiredness	17	(9.76%)	1	(5.88%)	1	(5.88%)	2	(11.76%)	13	(76.47%)
Fainting	8	(3.66%)	1	(12.50%)	2	(25.00%)	2	(25.00%)	3	(37.50%)
Edema	3	(3.66%)	0	(0.00%)	2	(66.67%)	0	(0.00%)	1	(33.33%)
Cough	3	(31.71%)	1	(33.33%)	0	(0.00%)	0	(0.00%)	2	(66.67%)

**Table 3 jcm-11-05276-t003:** The first step in the diagnostic process. Data are additionally stratified by age and presented as the number of patients who declared a particular step as the first during the diagnostic process and the percentage of the whole population or a particular subgroup. Abbreviations: GP, general practitioner.

First Diagnostic Step	Whole Population		30–40 Years		40–50 Years		50–60 Years		>60 Years	
	*n* of Patients	% of the Group	*n* of Patients	% of the Subgroup	*n* of Patients	% of the Subgroup	*n* of Patients	% of the Subgroup	*n* of Patients	% of the Subgroup
Reimbursed examination by the GP	45	(54.88%)	3	(6.67%)	5	(11.11%)	6	(13.33%)	31	(68.89%)
Examination by the cardiologist	18	(21.95%)	2	(11.11%)	2	(11.11%)	2	(11.11%)	12	(66.67%)
Calling an ambulance	9	(10.98%)	0	(0.00%)	0	(0.00%)	2	(22.22%)	7	(77.78%)
Appointment at the hospital	6	(7.32%)	0	(0.00%)	1	(16.67%)	1	(16.67%)	4	(66.67%)
Paid examination by the GP	4	(4.88%)	0	(0.00%)	0	(0.00%)	1	(25.00%)	3	(75.00%)

## Data Availability

The data presented in this study are available on request from the corresponding author. The data are not publicly available due to not obtaining consent from respondents to publish the data.

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
