# Peer review of "The Path of a Cardiac Patient—From the First Symptoms to Diagnosis to Treatment: Experiences from the Tertiary Care Center in Poland"

_jcm, 2022, doi:10.3390/jcm11185276_

Round 1
Reviewer 1 Report
This study provides a data about patients’ perception of diagnostic and treatment of CVD process. Results might provide an important clue to improve a public health care system. However, there are some questions to be addressed before publication.
One of the goals of this study was “to assess leading patient problems related to symptoms, diagnosis, and treatment of CVDs”. Authors present original questionnaire measuring different aspects of life. Is there any correlation between answers for these 27 questions and other questionnaires validated for QoL measurements (like WHOQOL-100 and the WHOQOL-BREF)? Weather this questionnaire was validated in any study with control healthy individuals?
Authors postulated that patients might have different perception of drug availability, quality of doctors’ appointments and effectiveness of treatment in different age. To support this conclusion patients of different ages should be adjusted by diagnosis and diagnostic procedures, but it was not clearly specified in the text. Spectrum of diagnosis, optimal (guidelines-based) and average (“real world”) time per diagnosis in all age groups should be specified.
Again, regarding cost-perception analysis it would be reasonable to understand if patients have different perception of the “same-cost-of-treatment” availability. The difference found in this study might be explain by significant difference of the cost and efficiency of treatment for different groups of diagnosis rather than age by itself.
Minor concerns
Table 1. Heart failure is not a diagnosis, but a complex syndrome caused by many diseases in different age groups. Diagnoses (conditions) require specification.
Line 131: the term “popular” is suboptimal in this context. Expressions “common”, “frequent” sound better here.
Reviewer 2 Report
Thank you for giving me the opportunity to review the manuscript titled "the path of a cardiac patient – from the first symptoms, by diagnosis, to treatment. Experiences from the tertiary care center in Poland."
In this the study authors aimed to assist the leading patient problems related to symptoms, diagnosis and treatment of cardiovascular diseases and to examine the patient's opinion about the healthcare system in the country. They created a 27 point, questionnaire and distributed it to 132 patients over a three month period out of which 82 patients were included in the study. The most common the cardiovascular diseases identified were the red man hypertension. They also identified the certain areas that needed the improvement including waiting time.
Introduction
Reasonably well written. Authors have identified the objective of the study clearly and have not overstated.
Material and methods
(1) authors excluded the survey responses which were not 100% completed. Excluding survey responses with incomplete responses is not as simple. There is generally a lot of information that is missed this way. For example, some patients would not like to answer a particular question because of sensitivity. Generally speaking, a missing data analysis should be performed
(2) authors mentioned that the responses were anonymous. However, they also mentioned that the "patient were noted to ensure that each questionnaire was completed by a different person.". How did they ensure that same person did not complete to questionnaire without including some form of identifiers?
(3) it seems like the authors designed their questionnaire. No information is given about how did they validate this questionnaire or pretesting.
Results
Too much information is given in the results, which can be overwhelming and it is hard to separate which is important and which is not. This can be trimmed down a little bit
Figures
Figure number one is big and bulky and the only thing it provides is the age breakdown. I do not think it provides any additional value to the paper
Figure number three. It is very hard to assess what this figure is depicting. This needs to be redrawn
Open questions
Open-ended questions need to be analyzed using qualitative research methodology which may provide additional insights
Overall this is a reasonable study. Although I am not sure about the external utility of this information. For example, in United States a lot of this information is already covered in the CMS mandated HCAHPS survey. It may be more useful locally and regionally to improve the quality of care provided at the hospital
Round 2
Reviewer 1 Report
All questions raised were properly addressed.